The effect of hepatopancreas homogenate of the Red king crab on HA-based filler

Ponomareva Tatyana 1
http://orcid.org/0000-0003-3776-5958 Sliadovskii Dmitrii 1
Timchenko Maria 1
Molchanov Maxim 2
Timchenko Alexander 3
http://orcid.org/0000-0001-7814-4563 Sogorin Evgeny 1 evgenysogorin@gmail.com
1 Federal Research Center “Pushchino Scientific Center for Biological Research of the RAS” , Pushchino , Russia
2 Institute of Theoretical and Experimental Biophysics of the RAS , Pushchino , Russia
3 Institute of Protein Research of the RAS , Pushchino , Russia
Uversky Vladimir
Electronic publication date: 2020 Feb 12
Publication date: 2020
Volume: 8
Electronic Location ID: e8579
Received 2019 Nov 22; Accepted 2020 Jan 16
Copyright: © 2020 Ponomareva et al.
Copyright year: 2020
Copyright holder: Ponomareva et al.
License: This is an open access article distributed under the terms of the Creative Commons Attribution License, which permits unrestricted use, distribution, reproduction and adaptation in any medium and for any purpose provided that it is properly attributed. For attribution, the original author(s), title, publication source (PeerJ) and either DOI or URL of the article must be cited.
License URL: https://creativecommons.org/licenses/by/4.0/

Keywords: Treating filler complications, Hyaluronidase, Red king crab, Hepatopancreas, Hyaluronic acid, Hyaluronidase activity, Turbidimetric method, Atomic force microscopy, Nuclear magnetic resonance

Funding: The authors received no funding for this work.

==============================
In this study, several methods were used to analyze the hydrolysis of hyaluronic acid (HA)-based cosmetic fillers by the hepatopancreas homogenate of the Red king crab. The results show that the homogenate and commercially available hyaluronidases have similar hydrolysis activities on the fillers. Atomic force microscopy images reveal that the HA fillers consist mainly of spherical-like particles, which are converted into filamentous structures as a result of hydrolysis by the Red king crab hepatopancreas homogenate. Turbidimetric analysis of the hydrolysis process shows that HA aggregation with acidic albumin exhibits a bell-shaped dependence on reaction time. Analysis of the hydrolysis process by nuclear magnetic resonance shows that HA degradation lasts several days. The maximum rate of the reaction is detected in the 1st h of incubation. The data confirm that the purified homogenate of the Red king crab hepatopancreas exerts hyaluronidase activity on HA-based cosmetic fillers; therefore, it may be considered as a potential therapeutic agent for treating filler complications.

Introduction

The increasing popularity and relative affordability of contour plastic using hyaluronic acid (HA)-based fillers have been associated with an increasing number of cases with adverse effects and hypercorrection (Park et al., 2011). The main complaints of patients after contour plastic procedures are dissatisfaction with the aesthetic appearance, swelling, and hypercorrection. HA injection may trigger acute inflammation, which requires rapid intervention and the removal of HA. Therefore, the development of a safe and effective method to eliminate the negative effects of HA injections is an urgent problem in cosmetology and medicine. Moreover, an increased concentration of HA around cancer cells has been shown to increase their resistance to the immune system and positively influence the ability of cancer cells to migrate, thereby increasing the probability of metastatic spreading (Knudson, 1996). As a consequence, the use of hyaluronidases in the treatment of cancer can be crucial.

Hyaluronic acid is a high molecular weight, negatively charged polymer. As a glycosaminoglycan, it consists of repeating disaccharide units (d-glucuronic acid and N-acetyl-d-glucosamine) linked by β-1, 4 and β-1, 3 glycosidic bonds. Depending on the source of HA, its disaccharide units can be repeated 2,000–25,000 times (106–107 Da). In an aqueous environment, HA has complex secondary and tertiary structures (Scott et al., 1991). Electron microscopy of HA reveals a cluster of interwoven filaments, which can intertwine, or star-shaped leaf formations (Gross, 1948; Scott et al., 1990). It forms loops and a loose glomus. Similar results have been obtained by Atomic force microscopy (AFM) (Cowman, Li & Balazs, 1998). HA, the main component of synovial fluid and the extracellular matrix, is involved in cell proliferation, differentiation, and migration, as well as water balance regulation (Ghosh, 1994).

Because of the chemical properties of HA, numerous approaches have been developed for its chemical modification. Industrial-scale methods have contributed to the emergence of a number of HA dosage forms that have been successfully used in medicine and cosmetology (Tiwari & Bahadur, 2019). Most commercially available cosmetic fillers contain both high molecular weight (about 1 MDa or higher) and low molecular weight (several hundred kDa) HA molecules. High molecular weight HA molecules often contain transverse covalent cross-links between each other, so they are less susceptible to degradation by endogenous hyaluronidases (Sall & Férard, 2007). In contrast, low molecular weight forms of HA do not form cross-links and provide lubrication during injections. The ratio of the two forms varies among products (Table S1).

To date, the direct injection of hyaluronidase into the HA-affected area has been the most reliable and painless approach to managing its adverse effects. This treatment is effective when the enzyme degrades the polymeric form of HA into its monomers. However, HA-based fillers produced by different manufacturers are degraded with varying efficiencies by hyaluronidase. In vitro studies have shown differences in the degradation of cosmetic HA preparations that contain different sources of hyaluronidase (from bovine or ovine testis). Thus, determining the correct dosage of hyaluronidase can present significant challenges (Sall & Férard, 2007; Flynn, Thompson & Hyun, 2013; Jones, Tezel & Borrell, 2010).

Hyaluronidase activity has been detected in the enzyme complex of the hepatopancreas homogenate of the Red king crab Paralithodes camtschaticus (order Decapoda) (Turkovsky et al., 2008). Immediately after incubation with the hepatopancreas collagenolytic protease complex, the molecular weight of HA was estimated using the capillary viscometer method. The authors proposed that the hyaluronidase activity is exerted by some (or several) of the nine proteases in the complex. Indeed, enzymes with broad specificity have been described in several studies. For example, some lipase and papain enzymes hydrolyze chitosan (Muzzarelli et al., 1995, 2002). It is likely that the hepatopancreas of the Red king crab (HPC) contains hyaluronidase, in addition to these proteases. Representatives of the Malacostraca class (e.g., the Norway lobster Nephrops norvegicus or Antarctic krill Euphausia superba), which includes the Red king crab, contain hyaluronidases in their hepatopancreas, and the biochemical properties of these enzymes have been characterized (Krishnapillai et al., 1999; Karlstam & Ljunglöf, 1991). Hepatopancreas hyaluronidase activity has also been found in some shrimps (Rosario & Nooralabettu, 2018). To our knowledge, the structures of the hyaluronidases in these animals have not yet been studied. The GenBank database contains only one predicted hyaluronidase gene of a representative of the order Decapoda, and it is from the shrimp Penaeus vannamei (NCBI Reference Sequence: XM_027376109.1). In summary, commercial crustacean species are promising sources of hyaluronidase products for use in cosmetology and medicine. Since the HPC is commercial waste, it is available as a raw material for large-scale processing to obtain a cosmetic hyaluronidase product.

In this study, the turbidimetric method, nuclear magnetic resonance (NMR) and AFM were used to investigate the degradation of HA fillers by commercially available hyaluronidases and hyaluronidase from the Red king crab hepatopancreas. The results of this work show that the studied hyaluronidases are comparably effective and, consequently, reveal opportunities to create a new hyaluronidase product from the HPC. In addition, the AFM and NMR results provide insights into the structural transformations of filler HA after its exposure to the hyaluronidase of the Red king crab hepatopancreas.

Materials and Methods

The HA products used in this study were Revofil Ultra (lot RTU-13-08), Teosyal Ultra Deep (lot TSU 110503B), Hyaluform filler deep (lot E0102 04/19), and HA sodium salt from rooster comb (H5388; Sigma–Aldrich, St. Louis, MO, USA). All tested fillers contain cross-linked and free HA in different ratios and different quantities of HA per one mL (Table S1; e.g., the Teosyal Ultra Deep filler stock preparation has 25 mg/mL HA). The HA fillers were stored in a place protected from light exposure at a temperature not exceeding 25 °C, and the prepared HA filler solutions were stored at 4 °C. The hyaluronidase products were Lidase (lot 110319) and Liporase (lot 080521). Both preparations were of animal origin and had different units of activity per 1 g of dry substance (Table S2). The hyaluronidase products were stored at 4 °C, and the prepared hyaluronidase solutions were stored at −20 °C.

The tested products (i.e., fillers and hyaluronidases) were purchased at a pharmacy.

Determination of hyaluronidase activity by the turbidimetric method

Turbidimetric analysis was conducted according to previously developed protocols, with minor changes (Dorfman & Ott, 1948; Rapport, Meyer & Linker, 1950; Tam & Chan, 1983). The content of the fillers was diluted by phosphate buffer (160 mM disodium phosphate and 39 mM sodium chloride; adjusted to pH 5.55 with hydrochloric acid) to a concentration of 2.5 mg/mL HA. An aliquot of the hyaluronidase solution was added to 0.2 mL of the HA solution, and the reaction mixture was thoroughly mixed and incubated at 37 °C. At certain time points during incubation, 10–35 μL aliquots were taken and then immediately mixed with phosphate buffer (to 0.5 mL) and 2.5 mL of acidic albumin (24 mM sodium acetate, 79 mM acetic acid, 50 mM sodium chloride, 1% albumin, pH 3.5). The mixture was thoroughly mixed and incubated at room temperature for 50 min. The absorbance was then measured at 600 nm on a Hitachi U-2000 spectrophotometer.

Preparation of the homogenate of HPC

The HPC homogenate was prepared as follows. The Red king crab Paralithodes camtschaticus was caught by a crab catching vessel from the company CJSC Arcticservice in the Barents Sea during crab fishing season. The hepatopancreases of several crabs were separated and immediately frozen, stored, and transported at −25 °C. After the hepatopancreas was thawed at room temperature in laboratory conditions, distilled water and ice were added at a weight ratio of 1:8:2, and the mixture was stirred at a low speed for 60 min. The final mixture was placed in a separating column and left for 2 h at −14 °C. The aqueous phase of the homogenate was filtered using a hollow fiber ultrafiltration module “AP-3-300” (Scientific-Production Complex (SPC) Biotest, Russia). The color of the sterile solution was dark amber. This solution was concentrated using a hollow fiber ultrafiltration module “AP-3-1” (SPC Biotest, Russia). The protein concentration in obtained hepatopancreas homogenate samples was in the range of 2–3.5 mg/ml. An aliquot of the final (concentrated) product was stored at −20 °C. Its protein concentration was determined by a colorimetric method using the biuret reagent (Lovrien & Matulis, 1995). Albumin (lot 0205C125; Amresco, Fountain Pkwy, OH, USA) served as the calibration protein.

Purification of the HPC homogenate

The precipitation procedure was carried out at 4 °C. Ammonium sulfate (up to 50% saturation) was added to 1 mL of the homogenate to remove impurities from the obtained preparation. Ammonium sulfate was slowly added while stirring. After centrifugation for 10 min at 10,000×g and 4 °C, the precipitate was dissolved in 0.5 mL of phosphate buffer and dialyzed in the same buffer for 24 h at 4 °C. The obtained sample was centrifuged for 10 min at 10,000×g and 4 °C and then passed through a 0.22 μm filter (Millipore, Burlington, MA, USA). The obtained solution was colorless. The sample was analyzed by gel electrophoresis according to Laemmli (1970).

Atomic force microscopy

The samples were prepared for AFM as follows. Buffer solution was used to bring the filler concentration to 2.5 mg/mL. Before their use, filler samples were incubated for 40 min at 37 °C. For hydrolysates, an equal volume of the sample was taken from the reaction mixture after 5, 40 and 120 min of incubation at 37 °C. We also analyzed the purified sample of the HPC homogenate diluted 21 times by phosphate buffer after it was incubated for 40 min at 37 °C. Next, two μL of the sample was transferred to freshly cleaved mica and incubated for 5 min. The sample was then washed twice in a drop of distilled water (deionized by a type I Milli-Q system) for 30 s and air-dried. AFM imaging was performed with an AFM Ntegra-Vita microscope (NT-MDT, Russia) in noncontact (tapping) mode in air. The typical scan rate was 0.5–1 Hz. Measurements were carried out using cantilevers NSG03 with a resonance frequency of 47–150 kHz, ensuring a 10 nm tip curvature radius. The processing and presentation of the AFM images were performed using Nova software (NT-MDT, Russia) and Gwyddion 2.44 software (http://gwyddion.net/, Czech Republic).

Nuclear magnetic resonance

One-dimensional (1D) 1H-NMR spectra were acquired with a Bruker Avance III 600 spectrometer (The Core Facilities Centre of Institute of Theoretical and Experimental Biophysics of the RAS) operating at a frequency of 600 MHz (1H) and a probe temperature of 310 K. The samples were placed into NMR tubes with a 5 mm diameter. Standard pulse sequences from the Bruker NMR pulse sequence library were used in the experiments. The 1D-pulse sequence ZGPR was applied for the suppression of proton signals from water. The free induction decay was collected into 96 K data points using an acquisition time of 3.42 s. The spectrum width was 24 ppm, the 90° pulse width was 16 μs, and the relaxation delay was 10 s. The NMR spectra were obtained sequentially with time fixation.

Hyaluronic acid from rooster comb or filler content was diluted with phosphate buffer (160 mM disodium phosphate, 39 mM sodium chloride, pH 5.55) to a concentration of 2.5 mg/mL HA. D2O (30 μL) and 30 μL of the purified sample of the HPC homogenate were added to 600 μL of HA solution preheated at 37 °C for 1 h. The sample was stirred, centrifuged for 1 min at 15,000×g, and placed into an NMR tube. The time points started with the addition of the HPC homogenate. After 50 spectra, the sample was transferred to an Eppendorf tube, and 10 μL of 4 mM 3-trimethylsilyl (2,2,3,3-2H4) propionate (TSP) in D2O was added to the sample as a standard marker. The mixture was placed into an NMR tube, and the 1H spectra were recorded. The control sample (600 μL of phosphate buffer, 30 μL of D2O and 30 μL of the decontaminated HPC homogenate preparation) did not produce any signals, except for 1H of H2O (Fig. S1). The Bruker TOPSPIN program was used to process the spectra and calculate integrals. The concentrations of HA were calculated at each time point on the basis of the proton signals of the acetyl group of the flexible or compact parts of the HA chains. The concentration was calculated relative to the known concentration of the added TSP marker. The molar concentration of HA was estimated from its disaccharide units, namely, d-glucuronic acid and N-acetyl-d-glucosamine.

The exponential curve of Figure “Hydrolysis of Hyaluform HA by purified HPC homogenate: (b) accumulation kinetics of the 1H signal of the N-acetyl-d-glucosamine acetyl group of HA chains (integral under the peaks)” was approximated, as described by C=1.2096+2.347∗(1−exp(−0.876∗x))+1.8997∗(1−exp(−0.02724∗x))

The exponential curve of Figure “Hydrolysis of HA from rooster comb by purified HPC homogenate: (b) accumulation kinetics of the 1H signal of the N-acetyl-d-glucosamine acetyl group of HA chains (integral under the peaks)” was approximated, as described by C=1.8559+1.3706∗(1−exp(−2.1222∗x))+1.7783∗(1−exp(−0.2954∗x))

Results

Turbidimetric analysis of HA-based fillers

The turbidimetric method was used to determine hyaluronidase activity. In contrast to the method for measuring the HA solution viscosity, the turbidimetric method is less laborious and requires a smaller amount of substrate. This method is based on two properties of high molecular weight HA: it binds to albumin in acidic conditions, and it forms aggregates. Light scattering by the aggregates was determined by using the dynamic light scattering method or measuring the optical density of the sample. As soon as the high molecular weight forms of HA are transformed into low molecular weight forms (as a result of hydrolysis), aggregates can no longer form. This phenomenon underlies the determination of hyaluronidase activity by this method and forms the basis for kinetic studies. It has been implicitly shown that HA aggregates no longer form with acidic albumin when the average molecular weight of HA is 6–8 kDa (Rapport, Meyer & Linker, 1950).

Figure 1A demonstrates the concentration dependence of the optical density on the amount of HA registered by turbidimetric analysis. It is important to note that unmodified HA obtained from rooster comb binds to albumin much more effectively than HA from the other sources. In further experiments, the optical density values were considered to be correct if they were on a linear part of these curves. Figure 1B shows the hydrolysis of the native HA from rooster comb. The decreased ability of HA to form aggregates during incubation with the homogenate was used as an indicator of HA cleavage.

Figure 1 Turbidimetric analysis of HA from rooster comb, Revofil Ultra, Hyaluform and Teosyal Ultra fillers: (A) the dependence of optical density on the concentration of HA in rooster comb, Revofil Ultra, Hyaluform and Teosyal Ultra in turbidimetric analysis; (B) HA from rooster comb during hydrolysis by the HPC homogenate in turbidimetric analysis; (C) kinetics of the hydrolysis of HA from Revofil Ultra filler using the HPC homogenate in triplicate; (D) kinetics of the hydrolysis of HA from the Hyaluform filler using commercially available hyaluronidases and the HPC homogenate; (E) kinetics of the hydrolysis of HA from the Revofil Ultra filler using three different concentrations of the HPC homogenate protein; and (F) kinetics of the hydrolysis of HA from the Hyaluform filler using the HPC homogenate and its decontaminated sample (“decont. HPC”).

Turbidimetric analysis of the filler HA hydrolysates resulting from the homogenate of the Red king crab hepatopancreas (HPC) produced an unexpected result (Fig. 1C). Instead of the anticipated reduced capacity of HA to form aggregates with acidic albumin, the opposite effect was observed in the first 40 min: the hydrolysis products of the filler HA bound to the protein and formed aggregates with it more effectively (i.e., the optical density increased). After 40 min of the reaction, the optical density in the turbidimetric samples decreased. The figure also demonstrates a sufficient level of reproducibility in the turbidimetric data. The same results for the hydrolyses of HA from Hyaluform (Fig. 1D) and Revofil Ultra (Fig. S2) were obtained with the commercially available hyaluronidases Lydase and Liporase.

The dependence of the hydrolytic efficiency on the total HPC homogenate protein in the reaction mixture was also investigated (see Fig. 1E). An increase in enzyme concentration resulted in the hydrolysis of HA into fragments that were unable to bind rapidly with acidic albumin. At a ratio of 1:1 (by weight) of total HPC homogenate protein to HA, little aggregation was detected after 10 min of hydrolysis, indicating that almost all HA molecules were converted into their short forms within 10 min. With a decrease in enzyme concentration in the reaction mixture, the conversion of HA to these short forms was slower. The most effective binding of acidic albumin to HA occurred (i) 10 min after the start of hydrolysis at a ratio of protein to HA of 1:5 and (ii) 20 min after the start of hydrolysis at a ratio of protein to HA of 1:10. Thus, a ratio of protein to HA of 1:10 or 1:15 was chosen as optimal for the experiments.

The hydrolysis reaction was stopped by increasing the temperature of the reaction mixture to 100 °C for 10 min: after 40 min of incubation at 37 °C, the reaction mixture (in a tightly closed Eppendorf tube) was placed in a boiling water bath for 10 min, after which the incubation was continued at room temperature (Fig. S3). This simple approach will make it easy to obtain partially hydrolyzed cross-linked HA in future studies (see “Discussion”).

AFM of HA-based fillers

Studying the HA hydrolysates using AFM required the purification of the HPC homogenate sample to remove pigments and contaminants (see “Materials and Methods”). The level of hyaluronidase activity of the purified homogenate was the same as that of the initial sample (Fig. 1F), and SDS-PAGE protein electrophoresis did not reveal significant differences in the sets of proteins between these two samples of the preparation (Fig. S4).

The AFM images of the HPC homogenate diluted 21 times by phosphate buffer show the presence of amorphous structures, but the decontaminated homogenate resembles “globular”-like structures with a height of 4–30 nm (Fig. S5). These are most likely complexes of proteolytic and other enzymes.

Since the filler HA solutions were stored at 4 °C, the samples were first warmed at 37 °C for 40 min before AFM analysis. The heat treatment did not significantly affect the results of the turbidimetric analysis (Fig. S6). The AFM images of the Hyaluform filler show spherical-like structures from several to 300 nm in height (Fig. 2).

Figure 2 AFM images (obtained in tapping mode) of the Hyaluform HA-based filler: (A) 50 × 50 μm field; (B) 1.4 × 1.4 μm field.

The population of these spherical-like structures can be divided into three groups according to their size and quantity. The first group includes a small population of structures 150–300 nm in height (Fig. 2A). Similar structures are observed in the Revofil Ultra filler preparation (Fig. S7). The second group is represented by a large population of structures 10–20 nm in height. An even larger population of structures are several nanometers in height and form the third group; these structures are formed as a dense layer that surrounds the other structures mentioned above (Fig. 2B).

The AFM method was used to analyze the hydrolysates of HA from the Hyaluform filler. The hydrolysates were obtained with the decontaminated HPC homogenate. Aliquots were taken from the reaction mixture at 5, 40 and 120 min of incubation and transferred to mica for AFM studies (Fig. 3; Fig. S8).

Figure 3 AFM images (in tapping mode) of the hydrolysates of the Hyaluform HA-based filler: (A) 5 min, (B) 40 min and (C) 120 min of HA hydrolysis with the HPC homogenate.

The AFM results show small gaps, consisting of spherical-like structures, on the sample surface (Fig. 3A) at the 5 min time point of hydrolysis. After 40 min of hydrolysis, these gaps are expanded (Fig. 3B). After 120 min, the spherical-like structures almost disappear, and filamentous mesh structures appear (Fig. 3C). The filamentous mesh structures are similar to those in the AFM images of non-cross-linked HA, as shown in (Jacoboni et al., 1999).

Furthermore, the AFM image of the reaction mixture after 5 min of incubation shows spherical-like structures about 600 nm in height or greater (Fig. S9). After 40 min of hydrolysis, there are fewer of these large structures, and many spherical-like structures up to 150 nm in height are observed. After 120 min, amorphous structures are detected in the samples, while structures up to 80 nm are practically undetectable.

The AFM analysis of native HA from rooster have shown the presence of an amorphous layer formed by aggregates up to 10 nm high, and a large number of globular-like aggregates up to 100–200 nm (Figs. S10 and S11). The hydrolysis of HA from rooster comb by purified HPC homogenate resulted in the disappearance of the most of large aggregates, the appearance of extended filaments and destruction of the amorphous layer with formation of various small aggregates (40 min of incubation). After 120 min of hydrolysis, the large aggregates almost completely disappeared and mostly small aggregates up to 8 nm high are present in the reaction mixture.

NMR of HA-based filler

The NMR spectrum of HA from Hyaluform reveals a weak 1H signal of the sugar rings in the range of 5.9–3.3 ppm and a minor 1H signal of the acetyl groups of N-acetyl-d-glucosamine at around 2 ppm (Fig. 4). The low detection levels of these signals are due to the close packing of the high molecular weight cross-linked molecules of HA in the filler.

Figure 4 1H-NMR spectra of the Hyaluform filler and its hydrolysis products after treatment by purified HPC homogenate.

The black curve represents the Hyaluform filler, and the red curve represents the hydrolysis products after 5 days of incubation.

The reaction mixture obtained from the hydrolysis of the Hyaluform filler by purified HPC homogenate was incubated in the NMR spectrometer at 37 °C for several days with NMR analysis at certain time points. Figure 4 shows a scan taken on the 5th day of sample incubation. The protons (1H) of the NH group, sugar rings, and acetyl groups of N-acetyl-d-glucosamine are observed to have significantly higher signals compared with the initial HA. No significant proton signals of acetic acid, amino acids, or other compounds are detected in the spectrum (the presence of these signals would indicate bacterial contamination of the sample during the incubation). Figure 5A shows details of the changes in the proton spectra of the acetyl groups of N-acetyl-d-glucosamine during the experiments. The protons of the acetyl group of the flexible parts of the HA chains are represented by a symmetric, narrow signal in the NMR spectrum, while the protons of the same acetyl group of HA included in associations (the compact part of HA chains) are represented by a wide, asymmetric signal that shifted to a high field by 0.04 ppm. The asymmetry of the peak is due to the different sizes of HA associations in the solution.

Figure 5 Hydrolysis of Hyaluform HA by purified HPC homogenate: (A) changes in the 1H signal of the N-acetyl-d-glucosamine acetyl group of HA chains in 1H-NMR spectra; (B) accumulation kinetics of the 1H signal of the N-acetyl-d-glucosamine acetyl group of HA chains (integral under the peaks).

The green marks represent the signal from the flexible part of the HA chains; the yellow marks represent the signal from the compact part of the HA chains; and the blue marks represent the total signals from both the flexible and compact parts of the HA chains.

The concentrations at each time point of the reaction were calculated from two proton signals of the HA acetyl group in both the flexible and compact parts of the HA chains (Fig. 5B). According to the NMR analysis, HA degradation in the reaction mixture lasts for several days, but the maximum reaction rate is observed in the 1st h of incubation. There are several mechanisms of hydrolysis: cross-link hydrolysis, sugar chain hydrolysis, and both reactions. All of these pathways decrease the size of HA associations in solution and thus increase their flexibility. This is confirmed by an increase in the corresponding signals in the proton NMR spectra; the integrated areas under the signals correspond to the apparent concentration.

NMR analysis of the hydrolysis reaction was performed for native HA from rooster comb (Fig. S12). The kinetics of the 1H signal accumulation of the N-acetyl-d-glucosamine acetyl group during the hydrolysis reaction of native HA had the same pattern as that of filler HA (Fig. S13). Since native HA does not have cross-links, it can be affirmed that sugar chain hydrolysis occurred for filler HA.

The real concentration of HA from the filler prior to hydrolysis was calculated as the concentration of its disaccharide units (d-glucuronic acid and N-acetyl-d-glucosamine); this value was 6.89 mM without taking cross-links into account. According to the NMR data, the total apparent concentration of the acetyl groups from the flexible and compact parts of the HA chains was 5.92 mM at the end of the experiment. The NMR spectra also reveal that small sugars were not formed as a result of hydrolysis by the homogenate.

Discussion

The focus of this study is the hydrolysis of HA in cosmetic fillers and the kinetics of HA hydrolysis using the hepatopancreas homogenate of the Red king crab. The results of turbidimetric analysis show that the efficiency of aggregate formation with albumin has a bell-shaped dependence on reaction time. The ascending part of the turbidimetric curve can be explained by the accumulation of HA hydrolysis products in the reaction mixture, which are more effective in forming aggregates with albumin than initial HA. The descending part of the curve is attributed to the degradation of HA molecules to short forms, which are unable to form aggregates with albumin. Most likely, the main reason for the difference in the form of the turbidimetric curve of the filler and native HA is the presence of cross-links in the filler, so that associates with albumin in the case of filler and native HA can vary significantly.

The AFM results show that HA in the fillers has a spherical-like structure, with a size that ranges from several to several hundred nanometers. Obviously, a spherical-like structural organization should be more resistant than non-modified HA to the action of hyaluronidase because only the surface molecules of HA in this structure are available for hydrolysis, while the molecules inside are protected from the enzymes. Under the action of hyaluronidase in the HPC homogenate, these structures became filamentous mesh structures.

The observed large spherical structures, with a height of approximately 600 nm (Fig. S9), can be explained by the association between proteins in the HPC homogenate and HA in the reaction mixture. Flynn, Thompson & Hyun (2013) investigated the kinetics of changes in the molecular weight of HA in the fillers Belotero Balance, Restylane, and Juvederm Ultra during a hydrolysis reaction using ovine testicular hyaluronidase. The results of liquid chromatography showed an increase in the molecular weight of HA in the first 30 min of the reaction. The authors explained this phenomenon as the result of the formation of enzyme–substrate complexes during the reaction. The enlarged spherical-like structures that we observed in the first 5–40 min of the reaction in our study and the detected increase in the molecular weight of HA in the work of Flynn, Thompson & Hyun (2013) probably share the same physical basis.

The 1H signal accumulation of the acetyl group of N-acetyl-d-glucosamine indicates the kinetics of enzyme activity by the HPC homogenate. These results show that hydrolysis lasted for several days, while the turbidimetric analysis curve reaches a plateau in 3 h under the same experimental conditions. This difference is attributed to a limitation of the turbidimetric assay, that is, the formation of aggregates is very sensitive to the size of HA.

In the first approximation, the binding of HA to acidic albumin may be considered a model for the formation of HA aggregates with proteins, which is normal in organisms. It follows that the partially hydrolyzed cross-linked HA is a potentially new form of HA, which combines the two properties necessary for its administration to joint cavities and the periarticular space for locomotor apparatus injuries and the treatment of arthritis and osteoarthrosis. It is essential that the effect of HA in therapy is prolonged for such diseases; moreover, the administered HA has to efficiently bind to protein molecules. Partially hydrolyzed cross-linked HA meets these requirements—it is more resistant to endogenous hyaluronidases than non-cross-linked HA, and it also efficiently forms aggregates with protein. Additional research is needed to support this hypothesis; however, even the simple thermal processing of the reaction mixture makes it easier to obtain samples of HA at different degrees of hydrolysis.

When the total protein concentration of the compared hyaluronidase preparations was equalized in the reaction mixtures, the HPC homogenate demonstrated hyaluronidase activity that was comparable to that of the commercially available products tested. Thus, this paper is the first report that demonstrates the prospects of using the HPC as a resource to create new hyaluronidase products for treating filler complications.

Products obtained from mammalian tissues are potentially dangerous to the consumer if the animal has spongiform encephalopathy because prion proteins from such animals can lead to Creutzfeldt–Jakob disease in humans. To date, this route of infection (through a hyaluronidase preparation) has not been described, but the state regulations of many countries prohibit the import and sale of these products in their territory. An exception is hyaluronidase preparations obtained from testis tissues of bulls grown in Australia. A quarantine regime of animal importation has been introduced in this territory, and it is believed that the local cattle population has never been exposed to spongiform encephalopathy. In this regard, the preparation of the Red king crab hepatopancreas hyaluronidase has a significant advantage since it does not pose this threat.

The main problem in the commercial-scale production of hyaluronidases from the Red king crab hepatopancreas is its difficult removal from the organism, as well as its storage and transportation in complicated marine fishing conditions. Solving these problems and conducting pre-clinical studies in an animal model to demonstrate the safety and therapeutic efficiency of the Red king crab hepatopancreas hyaluronidase may enhance the arsenal of preparations for treating filler complications in the near future.

Conclusions

In summary, the homogenate of HPC revealed the high hyaluronidase activity both in relation to native HA and in relation to cross-linked HA of cosmetic fillers. It is interesting that the turbidimetric curve for hydrolysis of HA from cosmetic fillers by hepatopancreas homogenate is bell-shaped. The hyaluronidase activity of HPC homogenate was comparable to that of the commercially available hyaluronidases. Thus, the HPC homogenate can be used as a basis for creation of new hyaluronidase products for treating filler complications. Our future studies will focus on isolation and detailed study of hyaluronidase from Red king crab hepatopancreas homogenate.

Supplemental Information

Supplemental Information 1 Experiment details of NMR analysis of native HA.

Data of half-time of spectrum registration of NMR, № spectrum, time of hydrolysis, and multiplicator to plot spectra to obtain Fig. S13.

Click here for additional data file.

Supplemental Information 2 Experiment details of NMR analysis of filler HA.

Data of half-time of spectrum registration of NMR, № spectrum, time of hydrolysis, and multiplicator to plot spectra to obtain Fig. 5.

Click here for additional data file.

Supplemental Information 3 Raw data exported from the Bruker Avance III 600 spectrometer applied for data analyses and preparation for the detailed investigation shown Fig. 5.

Raw data from NMR spectra of filler HA experiment. To plot the spectra use the multiplicator from Table S1. Spectral Region: LEFT = 9.001772643541111 ppm. RIGHT = 0.5040992680383022 ppm.

Click here for additional data file.

Supplemental Information 4 Raw data exported from the Bruker Avance III 600 spectrometer applied for data analyses and preparation for the detailed investigation shown Fig. S13.

Raw data from NMR spectra of native HA experiment. To plot the spectra use the multiplicator from Table S2. Spectral Region: LEFT = 2.1000129488236787 ppm. RIGHT = 2.0000032089015947 ppm.

Click here for additional data file.

Supplemental Information 5 Supplemental tables and figures.

Click here for additional data file.

The authors would like to thank Pozdnyakov Nikita for his technical support; Elena Demina and Sergei Lapaev for support in preparing this manuscript; Vyacheslav Sova for providing some reagent; Azat Abdulatypov for discussing the results; and Gennady Enin for encouragement.

Additional Information and Declarations

Competing Interests

Author Contributions

Patent Disclosures

Data Availability

The authors declare that they have no competing interests.

Tatyana Ponomareva conceived and designed the experiments, authored or reviewed drafts of the paper, and approved the final draft.

Dmitrii Sliadovskii performed the experiments, prepared figures and/or tables, and approved the final draft.

Maria Timchenko conceived and designed the experiments, performed the experiments, analyzed the data, prepared figures and/or tables, authored or reviewed drafts of the paper, and approved the final draft.

Maxim Molchanov conceived and designed the experiments, performed the experiments, analyzed the data, prepared figures and/or tables, and approved the final draft.

Alexander Timchenko analyzed the data, authored or reviewed drafts of the paper, and approved the final draft.

Evgeny Sogorin conceived and designed the experiments, prepared figures and/or tables, authored or reviewed drafts of the paper, supervision, and approved the final draft.

The following patent dependencies were disclosed by the authors:

Lebreton, P.F. (2010). Hyaluronic acid-based gels including lidocaine.

Volkov, V.P., Zelenetskiy, A.N., Akopova, T.A., Khabarov, V.N., Selyanin, M.A., and Selyanina, O.N. (2007). Method for preparation of cured hyaluronic acid salts in water medium.

Simseongbo, Baksohyeon, Hongujin, and Choeeunho (2015). Structure in which active material is inserted into de-differentiated plant protoplast, method for preparing same, and cosmetic composition containing same.

Nizhechik, Y.S., Pestova, I.A., Ananicheva, L.L., and Tarasova, T.N. (2000). Method of high-quality preparation “Lidaza” preparing.

Nekrasov, A.V., Karapututze, T.M., Medvedev, S.A., Kozyukov, A.V., and Karapututze, N.T. (2017). Method for preparing conjugate of hyaluronidase with derivatives of polyethylene piperazine and application of produced conjugate.

The following information was supplied regarding data availability:

NMR raw data is available in the Supplemental Files.

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
