# Peer review of "The effect of hepatopancreas homogenate of the Red king crab on HA-based filler"

_PeerJ, doi:10.7717/peerj.8579_

## Round 0.1 · original submission · Minor Revisions

Please address all issues pointed by both reviewers and revise your manuscript accordingly.

Reviewer 1 ·

Basic reporting

no comment

Experimental design

no comment

Validity of the findings

no comment

Additional comments

In the manuscript (#43188) entitled “The effect of hepatopancreas homogenate of the Red king crab on HA-based filler”, the authors studied the hydrolysis of hyaluronic acid (HA) in cosmetic fillers and the kinetics of HA hydrolysis using the hepatopancreas homogenate of the Red king crab. The authors found that the hepatopancreas homogenate of the Red king crab exhibits comparable enzyme activity to commercially available hyalurionidases. Turbidimetric analysis of HA from fillers shows a bell-shaped dependence of aggregation with acidic albumin on reaction time, which is totally different from the turbidimetric analysis result of HA obtained from rooster comb. The authors then investigated the structural transformations of HA products during the hydrolysis process using AFM and NMR analysis. The study provides industry an opportunity to use the hepatopancreas of the Red King crab as raw materials for the commercial-scale production of hyaluronidases. It not only has medical advantages than the current commercially available hyaluronidases, mainly from bovin, but also has great economical benefit.

Moderate comments:

1. The authors gave a very simple explanation for the distinct turbidimetric analysis result of HA from cosmetic fillers than that from the rooster comb in the Discussion section. Considering that it is the first important finding of this manuscript, the authors should provide more evidence and analysis to prove their argument. Given the size/molecular weight (and morphology?) of HA regulates its binding to albumin and the forming of aggregates. AFM analysis of the rooster comb HA and the hydrolysates could be performed and compared with the AFM results with filler HA, which may give us some clues.

2. The current study still cannot make a conclusive determination of what contributes to the
hyaluronidase activity of the hepatopancreas homogenate. Is it hyaluronidase or other enzymes? If the authors can clarify it using other sophisticated analytical techniques like liquid chromatography and mass spectrometry, it will be a significant contribution to this area.

Minor comments:

Citation numbers are out of order throughout the text, and references are in different formats, should be cleaned up according to the requirement of the journal. Manually change them or use the correct bibliography style.

Line 96. “distilled water and ice were added in a ratio of 1:8:2”. Do you mean “distilled water and ice were added at a weight ratio of 1:8:2, hepatopancreas:water:ice, ” ?

Line 102. What is the protein concentration, in the range of *** ?

Line 111. For the gel electrophoresis analysis (Fig S3), the two commercial hyalurionidases products, Lidase and Liporase, should be run simultaneously. It will help you figure out which band of the hepatopancreas homogenate corresponds to the presumed hyalurionidase. And it is also a positive control for the detection of hyalurionidase in the homogenate.

Line 192. Figure S1 should be referenced before Figure S2 (Line 177). Numbered in order of first appearance in the text.

Reviewer 2 ·

Basic reporting

The authors characterized the hydrolysis of hyaluronic acid (HA)-based cosmetic fillers by the hepatopancreas homogenate of the Red king crab using several biophysical methods and showed that hepatopancreas homogenate of the Red king crab has hyaluronidase activity. The present work established the potential of commercial-scale production of hyaluronidases from the Red king crab hepatopancreas. The manuscript is written with clear English and professional organizations. The background shows proper context with relevant references. The first reference starts with number 15. The reference order needs to be corrected. The raw data were provided with the manuscript. The figures and tables are shown with good quality and support the hypotheses.

Experimental design

Research questions are well defined, relevant and meaningful. The methods are described in the manuscript in detail and are sufficient to follow.

Validity of the findings

The authors successfully characterized the hydrolysis of hyaluronic acid (HA)-based cosmetic fillers by the hepatopancreas homogenate of the Red king crab by turbidimetric analysis, AFM and NMR techniques. Data were interpreted completely and supported the conclusion that homogenate of the Red king crab hepatopancreas has hyaluronidase activity on HA-based cosmetic fillers.

Can you provide 1H-NMR spectra of native HA from rooster comb and its hydrolysis products after treatment by purified HPC homogenate as supplementary material? Please provide the overlay of 1H-NMR spectra of at different time points as shown in the article for Hyaluform HA and its hydrolysis products by purified HPC homogenate.

---

## Round 0.2 · accepted · Accept

All critiques of both reviewers were adequately addressed and the manuscript was revised accordingly.

Reviewer 1 ·

Basic reporting

no comment

Experimental design

no comment

Validity of the findings

no comment

Additional comments

Minor comment:
The number format in the graphic legend of Fig. 4 and Fig. 5(a), as well as in Figs. S12, S13, is not consistent (e.g., 114.22 h VS 114,22 h). The first format is better.

Reviewer 2 ·

Basic reporting

The authors satisfactorily addressed all the comments and provided the requested supplementary figures.

Experimental design

No comment

Validity of the findings

No comment